# Chronic Olfactory Dysfunction in Children with Long COVID: A Retrospective Study

**DOI:** 10.3390/children9081251

**Published:** 2022-08-19

**Authors:** Danilo Buonsenso, Laura Martino, Rosa Morello, Cristina De Rose, Piero Valentini

**Affiliations:** 1Department of Woman and Child Health and Public Health, Fondazione Policlinico Universitario A. Gemelli IRCCS, Rome 00168, Italy; 2Global Health Research Institute, Istituto di Igiene, Università Cattolica del Sacro Cuore, Rome 00168, Italy

**Keywords:** COVID-19, long COVID, children, anosmia

## Abstract

Olfactory dysfunction is one of the long-term consequences of acute SARS-CoV-2 infection in adults. This study aims to analyze the prevalence of chronic anosmia among COVID-19 children and to bring to light its impact on their families’ quality of life and wellbeing. Children younger than 18 years old, who were detected as being COVID-19-positive by RT-PCR and were assessed in a pediatric post-COVID outpatient clinic at least 28 days after the onset of the acute infection, were included in the study. The patients suffering from persisting smell disorders were asked to answer a questionnaire about their symptoms and how they influence their daily life. Out of the 784 children evaluated, 13 (1.7%) presented olfactory impairment at a mean follow-up since the acute infection of more than three months. Parents’ answers showed that they were worried about their children’s health, in particular they wanted to know if and when they would recover and if these disorders would have long-term consequences. They also wanted to share their experiences, in order to help other people who are experiencing the same disorders in everyday life. Our study highlights that smell disorders can significantly upset children’s eating habits and everyday activities. Furthermore, these findings suggest that future research should try to better understand the mechanisms causing loss of smell in COVID-19 patients and find the most appropriate treatment.

## 1. Introduction

Post-COVID-19 condition, or Long COVID, is now a well-established negative long-term outcome of SARS-CoV-2 infection in adults [1]. Despite initial uncertainties, there is now evidence that children can also develop persistent sequelae of COVID-19 as well as adults [2,3], although its real incidence is still unclear. While a definition of Long COVID in adults was relatively quickly released by the World Health Organization and adapted during the pandemic [4], only in March 2022 was a pediatric definition developed through a Delphi process involving both researchers and family advocates [5]. A child should be considered as having Long COVID if: (i) they have signs or symptoms that persist for a minimum duration of 12 weeks after initial testing for COVID-19 (even if symptoms have waxed and waned over that period); (ii) symptoms impact their physical, mental or social wellbeing and are interfering with some aspect of daily living (for example, school, work, home or relationships); (iii) symptoms have continued or developed after a diagnosis of COVID-19 (confirmed with one or more positive COVID tests); (iv) symptoms cannot be explained by other known conditions. Therefore, among these symptoms, persisting anosmia can be a new symptom that can last more than 12 weeks and impact on daily life. Currently, there is increasing evidence that persistent olfactory dysfunction is a frequent complication of acute SARS-CoV-2 infection in adults [6,7]. Several researchers have been evaluating the mechanisms causing loss of smell in COVID-19 patients, but they are not completely understood. Sensorineural inflammation of the olfactory neuroepithelium is one of the main events that is thought to be involved in the pathogenesis of anosmia. Recent molecular research showed that SARS-CoV-2 infection seems to induce the persistent rearrangement of neuronal nuclear architecture which results in the downregulation of olfactory receptor genes [8]. Furthermore, brain imaging studies reported a decreased metabolic activity of the amygdala, uncus, parahippocampal gyrus, the pons and cerebellum in patients with long-lasting hyposmia/anosmia [9,10]. However, systematic studies conducted on pediatric patients who suffer from persistent olfactory dysfunction after acute SARS-CoV-2 infection are still lacking. Olfactory impairment involves the inability to recognize odors, which may significantly affect personal hygiene. Patients with anosmia may report reduced appetite and food enjoyment, which could lead to decreased nutritional intake. Overall, the olfactory deficit could affect children and adolescents’ relationships with family and friends, resulting in increased social isolation and depression. Pediatricians need to define the proportion of young patients reporting persistent olfactory dysfunction, to study how they respond to it and if they change their eating habits and social life. Furthermore, some researchers have evaluated the efficacy of olfactory training and topical corticosteroids in the management of COVID-19-induced anosmia [11], but results are controversial, and more studies are needed to find the most appropriate therapeutic approach to rehabilitate the sense of smell.

This study aims to analyze the prevalence of long-lasting loss of smell among children and to evaluate its severity, how it influences their families’ quality of life and what we can do to help them to recover.

## 2. Materials and Methods

This is a retrospective study of children younger than 18 years of age with a microbiologically confirmed diagnosis (based on SARS-CoV-2 detected on a nasopharyngeal swab by RT-PCR) of SARS-CoV-2 infection that were assessed in a pediatric post-COVID outpatient clinic of Fondazione Policlinico Universitario A. Gemelli IRCSS in Rome, Italy. In our outpatient clinic, we evaluate children that either fully recovered from acute infection or present persisting symptoms. Children can be sent to the post-COVID unit either after discharge from our institution, or directly sent from the family pediatricians (and therefore not seen at baseline during acute infection). Data from all consecutive patients evaluated in our outpatient unit from 5 February 2021 to 1 June 2022 were included.

### 2.1. Inclusion and Exclusion Criteria

See Table 1 for the inclusion criteria used.

For each child, we collected the main demographic data, COVID-19 vaccination status, severity of acute disease according to the initial classification adapted according to our local practice (e.g., moderate disease only in case of symptomatic pneumonia documented by imaging, since we do not perform routine chest imaging in children without signs and/or symptoms of lower respiratory tract infections) [12], mean time of follow-up and other concomitant persisting symptoms.

Anosmia was defined as the loss of sense of smell, or reduced ability to smell, or distorted perception of selective smells that were normally perceived before COVID-19.

Moreover, in order to understand the impact of anosmia on daily life, we asked families of children with persistent anosmia the following questions in Table 2.

### 2.2. Statistical Analysis

Statistical analysis was performed using…

Categorical variables were presented as count and percentage; they were compared using Chi-squared tests or Fisher’s exact tests. Continuous variables were expressed as mean with standard deviation or median with interquartile range (IQR 25–75%), depending on the normality distribution. Mann–Whitney U test was used to compare continuous variables if not normally distributed. A *p* value < 0.05 was considered statistically significant.

### 2.3. Ethics Statement

The study was approved by the ethic committee of Fondazione Policlinico Universitario A. Gemelli IRCSS in Rome, Italy (ID 4518, Prot n 0040139/21, date 15 November 2021). Written informed consent was obtained from all participants or legal guardians for the participation in this study before beginning the study.

## 3. Results

### 3.1. Characteristics of the Population

A total of 784 children diagnosed with SARS-CoV-2 infection were enrolled in the study. The median age of our population was 7.99 years: 530/784 (67.6%) were aged 0 to 9 years and 254/784 (32.4%) were aged 10 to 18 years. Regarding sex distribution, 45.5% (357/784) were girls. Out of the infected children, 89 had known pre-existing comorbidities. The most noted pathologies were allergic asthma, autism spectrum disorders, allergies and atopic dermatitis. Only 73 patients (9.3%) completed the COVID-19 vaccination regimen before the infection, and 56 children (7.1%) received at least one dose of the vaccine. During the acute SARS-CoV-2 infection, 39 children (5.0%) were asymptomatic, 724 (92.3%) had mild disease, 19 (2.4%) had moderate disease and 2 patients (0.3%) had severe disease. Overall, 24 (3.1%) children were hospitalized and 4 (0.5%) needed pediatric intensive care unit admission. The median duration of the follow-up time since the initial diagnosis of SARS-CoV-2 infection was 106.9 days.

Among the children followed up in our outpatient clinic, 13/784 (1.7%) suffered from persistent olfactory dysfunction after acute SARS-CoV-2 infection, 771/784 (98.3%) did not complain about long-lasting smell disorders. The distribution of age, gender, nationality, comorbidities, COVID-19 vaccination status, acute disease severity, mean time of follow-up and post-acute infection symptoms are reported among the different groups (Table 3).

The median age of children suffering from persistent anosmia was 13.86 years: 12 of them (92.3%) were aged >10 years, 1 child was 9 years old and 8 of them (61.5%) were girls. The statistical analysis showed that the distribution of age between the anosmic and non-anosmic patients was significantly different (*p* value < 0.05). One of them had allergic asthma and two suffered from allergies. Only two of them were fully vaccinated before the SARS-CoV-2 infection. All of them had mild acute disease and did not require hospitalization. The median duration of the follow-up time from the COVID-19 diagnosis was 132 days. Seven children (53.8%) suffering from olfactory dysfunction complained about taste disorders too. The most common concomitant persisting symptoms among these children were dyspnea on exertion, headache, muscle and joint pain, asthenia and palpitations. We found that the presence of altered taste, dyspnea at rest, dyspnea on exertion, chest pain, palpitations and joint pain was significantly different between the two group of patients (*p* < 0.05). We summarize the clinical situation of each patient in Table 4. In these patients, the clinical presentation of a broad spectrum of symptoms, persisting after the acute SARS-CoV-2 infection, did not suggest that there were other underlying pathologies that could cause olfactory impairment and fit with the diagnosis of Long COVID. Durations of anosmia reported by the 13 children can be seen in Figure 1. The median duration of this symptom was 120 days.

### 3.2. Families’ Voices

Among the children with persistent anosmia, 11/13 (85%) reported having suffered a reduced or distorted ability to smell since the acute phase of SARS-CoV-2 infection. Only 2/13 (15%) started experiencing olfactory dysfunction two months after the positive COVID test. Clinical manifestations of smell disorders can vary widely. Most of our patients (10/13) reported the inability to detect any smell. In particular, four parents firstly noticed that their children were not able to recognize cooking scents or odors. As reported by families, anosmia had a mild-moderate impact on their daily routine, and it caused an excessive use of perfumes and deodorants in four of our patients. One of our patients, a 12-year-old adolescent who cannot smell anything, is looking forward to recovering and she is constantly asking her mother when she will be able to recognize the odors again.

Only one family reported that their child’s loss of smell had a slight impact on their daily activities. One patient told us that they were bothered by some smells. Two children reported a completely distorted perception of smell: for a 13-year-old adolescent, all smells were disgusting, cooking scents were not tolerated, and perfumes were not correctly perceived. Since this patient stopped eating her favorite food and lost weight, her mother told us that “her olfactory dysfunction has a devastating impact on the family’s quality of life”, they changed all their eating habits and she is worried because she does not know how to replace food that her child refuses. Another adolescent complained about an altered perception of smells as well. He said that he found human sweat disgusting. Both of them reported taste disorders as well, and they experienced these alterations two months after the acute infection. All the parents interviewed shared the same feeling—fear. They were worried about their children’s health; they want to know if and when they will recover and if these disorders will have long-term consequences. Our families trust the scientific community, encourage our clinical research and hope we will soon find an efficient treatment for anosmia. They also want to share their experiences, in order to help other people who are experiencing the same disorders in everyday life.

## 4. Discussion

In this study, we assessed the impact of chronic anosmia on a relatively large population of children assessed in a referral post-COVID Unit in Italy. Overall, we found that 1.7% of the assessed children still presented a smell problem at a mean follow-up of more than three months since the acute infection, and this problem had an impact on their habits and on family wellbeing. Older age and the persistence of other symptoms such as altered taste, dyspnea at rest, dyspnea on exertion, chest pain, palpitations and joint pain was significantly different between the two groups of patients, with it being more frequent in the group of chronic smell dysfunction. Specifically, nine out of the thirteen patients (69.2%) had the coexistence of multiple symptoms, making their picture more specifically related to Long COVID, while the others had either smell disjunction alone or associated taste problems.

The prevalence of chronic anosmia among children has not been established. To our knowledge, there is only one study [13], conducted in Turkey, that provided detailed clinical and demographic characteristics of two populations that either developed anosmia or did not. The study reported that in 8.4% of cases, anosmia did not regress one month after the infection. More importantly, although persistent anosmia has always been understated in previous discussions and publications, the interviews we conducted with children and parents highlight that this symptom may clearly impact the family routine, leading to changes in eating habits or in other social activities (e.g., sport or social relationships in children). The qualitative questionnaire we administered to the families allowed us to gain insight into their experiences. The answers showed that olfactory impairment mainly affects nutritional intake and personal hygiene, bringing to light the importance of the sense of smell in everyday life. As reported by parents, the inability to detect any smells reduced in some children the interest in food, leading them to change their food tastes and habits drastically. Because of the children’s refusal of some dishes, eating together as a family switched from an important moment of sharing and conversation to a tense moment, characterized by the struggles of their parents to put new and interesting food on the table. Another important effect reported in patients with olfactory impairment is their doubt regarding personal hygiene. Some of our patients worried about their own body odor which resulted in an excessive use of perfumes and deodorants. This aspect significantly affected the wellbeing of adolescents, who like to take care of their appearance, and their social relationships. Even more severe is the frustration of parents who cannot make their children feel better. This impact highlights the need to also understand an appropriate management of this condition. Although some researchers have evaluated the efficacy of olfactory training and topical corticosteroids in the management of COVID-19-induced anosmia [11], the results are controversial, and currently, there is a paucity of information on the most appropriate therapeutic approach to rehabilitate the sense of smell. A recent study [14] showed that the loss of mitochondrial membrane potential could be related to Long COVID, which may indirectly support the recent findings of pathological events happening at a subcellular level, including neuro-cellular derangement in animal models of anosmia [9]. These findings suggest that interventions using antioxidants and microelements conducted early in the course of the disease (including neuroprotectors) [15], along with olfactory training, may deserve to be studied in well-designed studies aimed at assessing these options in the management of chronic anosmia. Importantly, a recent editorial on Nature highlighted that how to recognize and treat Long COVID still represents an important gap in current medical knowledge [16]. Considering that a recent paper showed that millions of adults are living with long-lasting smell problems after SARS-CoV-2 infections [17], understanding the proper management of this condition should be a priority for researchers and health policy makers, in addition to the current need to better understand how we can manage acute SARS-CoV-2 infection in children. In fact, a recent review [18] confirmed the milder disease course of acute infection in children, and it also analyzed several different therapeutic options addressed in current literature (including antivirals, immunomodulation, aerosol, probiotics, lactoferrin, antibiotics and vaccines) without finding enough strong evidence to derive conclusions about the efficacy of these options on reducing both short- and long-term consequences of COVID-19 in children. Therefore, addressing how early interventions or prevention through vaccination impact short- and long-term quality of life in children with SARS-CoV-2 infection remains a priority.

Our study has limitations to address. Firstly, it is a retrospective study, and therefore, patients were not assessed at the same time of follow-up since the initial acute infection. Secondly, smell problems were defined according to a self-description of smell impairment by the patients, in the absence of a standardized diagnostic tool [19]. Thirdly, other investigations to address whether smell problems were due to other reasons were not performed (other than anterior rhinoscopy made during clinical examination), although it is very uncommon in pediatrics to find any alternative diagnosis causing long-lasting smell problems. Lastly, the questionnaire assessing the negative impact of smell problems in families with children with long-lasting smell problems is not yet validated, although we strongly believe that this preliminary pilot information will help the development of new tools to be used on larger populations.

In conclusion, we provided the first detailed characterization of a cohort of children that developed chronic anosmia after COVID-19 and its impact on daily life. These findings highlight the need to better understand why some young people develop this problem and what its best management is.

## Figures and Tables

**Figure 1 children-09-01251-f001:**
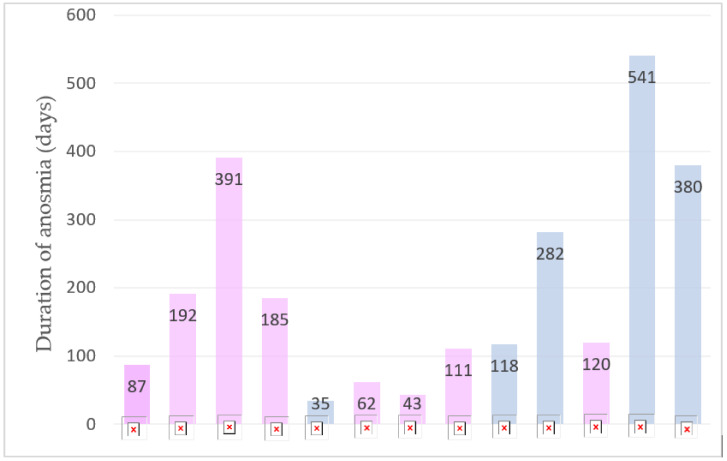
Duration of anosmia (days) in the 13 children.
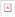
: It’s the patient number, 1-2-3-4 etc.

**Table 1 children-09-01251-t001:** Inclusion criteria used.

Children aged 0–18 years
The child sought/needed primary or secondary medical care for COVID-19
Laboratory (RT-PCR) diagnosis of acute COVID-19
At least 28 days from the onset of COVID-19 symptoms
Parent’s/carer’s/guardian’s consent to participate

**Table 2 children-09-01251-t002:** Questions asked to children’s families.

1	When was your child diagnosed with SARS-CoV-2 infection?
2	Did your child report smell disorders? If yes, since when?
3	Did your child report taste disorders? If yes, since when?
4	If your child had smell disorders, how long did they last?
5	If your child had taste disorders, how long did they last?
6	Could you tell us how your child describes smell and taste disorders?
7	Could you tell us how these disorders impact on your child’s daily routine?
8	As a parent, do these disorders impact on your daily routine or mood, too? If yes, describe us some examples.
9	As a parent, what do you expect from the scientific community and media about these problems?

**Table 3 children-09-01251-t003:** Demographic and clinical characteristics of the subject enrolled.

	All	Persistent Anosmia	Non-Persistent Anosmia	*p* Value
N (%)	784 (100)	13 (1.7)	771 (98.3)	
Age at first SARS-CoV-2 infection (years) N (%)				
0–9 years	530 (67.6)	1 (7.7)	529 (68.6)	
10–18 years	254 (32.4)	12 (92.3)	242 (31.4)	
Median age (years) IQR	7.99 4.6–10.8	13.86 11.6–14.5	7.83 4.55–10.7	<0.05
Gender N (%)				0.24
Male	427 (54.5)	5 (38.5)	422 (54.7)	
Female	357 (45.5)	8 (61.5)	349 (45.3)	
Nationality N (%)				<0.05
Italy	778 (99.2)	12 (92.3)	766 (99.4)	
Other countries	6 (0.8)	1 (7.7)	5 (0.6)	
COVID-19 Vaccination status N (%)				0.78
Non vaccinated	648 (82.7)	10 (76.9)	638 (82.7)	
Vaccinated with 1 dose	56 (7.1)	1 (7.7)	55 (7.1)	
Fully vaccinated	73 (9.3)	2 (15.4)	71 (9.2)	
Vaccinated with booster dose	7 (0.9)	0 (0)	7 (0.9)	
Comorbidities N (%)				0.18
Yes	89 (11.4)	3 (23.1)	86 (11.2)	
No	695 (88.6)	10 (76.9)	685 (88.8)	
Allergic Asthma	15 (1.9)	1 (7.7)	14 (1.8)	
Asthmatic Bronchitis	15 (1.9)	0	15 (1.9)	
Autism Spectrum disorders	12 (1.5)	0	12 (1.6)	
Allergies	8 (1.0)	2 (15.3)	6 (0.8)	
Atopic dermatitis	8 (1.0)	0	8 (1.0)	
Adenotonsillar hypertrophy	5 (0.6)	0	5 (0.6)	
Recurrent respiratory infections	4 (0.5)	0	4 (0.5)	
Prematurity	4 (0.5)	0	4 (0.5)	
Gastroesophageal reflux	2 (0.3)	0	2 (0.3)	
Celiac disease	2 (0.3)	0	2 (0.3)	
Epilepsy	2 (0.3)	0	2 (0.3)	
Migraine	2 (0.3)	0	2 (0.3)	
D. Duchenne	2 (0.3)	0	2 (0.3)	
Henoch–Schonlein Purpura	2 (0.3)	0	2 (0.3)	
Obesity	2 (0.3)	0	2 (0.3)	
Down syndrome	2 (0.3)	0	2 (0.3)	
Noonan syndrome	1 (0.1)	0	1 (0.1)	
Klinefelter syndrome	1 (0.1)	0	1 (0.1)	
Turner syndrome	1 (0.1)	0	1 (0.1)	
Arnold–Chiari malformation	1 (0.1)	0	1 (0.1)	
Charcot–Marie–Tooth syndrome	1 (0.1)	0	1 (0.1)	
Primary ciliary dyskinesia	1 (0.1)	0	1 (0.1)	
Solitary kidney	1 (0.1)	0	1 (0.1)	
No comorbidities	690 (88.0)	10 (77.0)	680 (88.2)	
Acute disease severity N (%)				0.78
Asymptomatic	39 (5.0)	0	39 (5.1)	
Mild	724 (92.3)	13 (100)	711 (92.2)	
Moderate	19 (2.4)	0	19 (2.5)	
Severe- MISC	2 (0.3)	0	2 (0.3)	
Hospital admission N (%)				0.33
Yes	24 (3.1)	1 (7.7)	23 (3.0)	
No	760 (96.9)	12 (92.3)	748 (97.0)	
PICU admission N(%)				0.79
Yes	4 (0.5)	0	4 (0.5)	
No	780 (99.5)	13 (100)	767 (99.5)	
FUP				
Mean (days) IQR	106.9 61–120	132.0 62–185	106.7 61–119	0.43
Post-acute infection symptoms				
Yes	287 (36.6)	13 (100)	274 (35.5)	
No	497 (63.4)	0	497 (64.5)	
Fever	11 (1.4)	0	11 (1.4)	0.66
Nasal congestion/rhinorrhea	25 (3.2)	1 (7.7)	24 (3.1)	0.35
Altered taste	11 (1.4)	7 (53.8)	4 (0.5)	<0.05
Cough	33 (4.2)	1 (7.7)	32 (4.2)	0.528
Dyspnea at rest	7 (0.9)	1 (7.7)	6 (0.8)	<0.05
Dyspnea on exertion	66 (8.4)	6 (46.2)	60 (7.8)	<0.05
Asthma	10 (1.3)	0	10 (1.3)	0.68
Chest pain	33 (4.2)	1 (7.7)	32 (4.2)	<0.05
Palpitations	21 (2.7)	2 (15.4)	19 (2.5)	<0.05
Joint pain	35 (4.5)	3 (15.4)	32 (4.2)	<0.05
Muscle pain	50 (6.4)	2 (15.4)	48 (6.2)	0.18
Headache	72 (9.2)	3 (23.1)	69 (8.9)	0.08
Asthenia	107 (13.6)	3 (23.1)	104 (13.5)	0.32
Gastrointestinal symptoms	48 (6.1)	1 (7.7)	47 (6.1)	0.81
Rash	18 (2.3)	1 (7.7)	17 (2.2)	0.19
Other: yes	75 (9.6)	3 (23.1)	72 (9.3)	0.09

**Table 4 children-09-01251-t004:** Clinical situation of patients suffering from persistent anosmia.

Patient 1	Anosmia, dyspnea on exertion, chest pain, palpitations, joint pain, rash.
Patient 2	Anosmia, altered taste, asthenia.
Patient 3	Anosmia, altered taste, dyspnea on exertion, palpitations.
Patient 4	Anosmia, concentration and memory disorders.
Patient 5	Anosmia, altered taste, joint and muscle pain, headache, gastrointestinal symptoms, asthenia.
Patient 6	Anosmia, dyspnea on exertion.
Patient 7	Anosmia, nasal congestion, dyspnea on exertion, muscle pain, asthenia.
Patient 8	Anosmia and altered taste.
Patient 9	Anosmia and altered taste.
Patient 10	Anosmia and altered taste.
Patient 11	Anosmia, altered taste, cough, dyspnea on exertion, headache.
Patient 12	Anosmia.
Patient 13	Anosmia, dyspnea at rest, dyspnea on exertion, chest pains, joint pain, headache, altered sleep–wake rhythm.

## Data Availability

Data are available upon request to the corresponding author.

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
