# Peer review of "Chronic Olfactory Dysfunction in Children with Long COVID: A Retrospective Study"

_children, 2022, doi:10.3390/children9081251_

Round 1

Reviewer 1 Report

Buonsenso and colleagues present a retrospective study assessing the prevalence of anosmia among children < 18 consulting for persisting symptoms related to COVID-19. The authors report the data from 784 children and teenager from a single center in Italy and find 1.7% of reported persistent anosmia.

The results are interesting although similar data have been reported. Several points need to be considered.

Major point :

- The introduction focuses on Long-COVID in children, but it is not clear if the included patients were satisfying the criteria for long-COVID (for other symptoms than anosmia).

- Another major point to discuss is that the population is a population (of parents) seeking medical attention after COVID-19. Since it is not a systematic analysis of infected patient the prevalence of anosmia cannot be generalized. Please discuss in the limitation.

- In the abstract the authors state that anosmia is a COMMON long-term consequence of SARS-CoV-2 in adults. This is fairly rare, from 3-8% (10.3390/pathogens11020269). The introduction should contain what is already known on the subjet in adults and more importantly in children. The what is the gap of knowledge and then what the authors want to address. 

- Even if there are few patients with anosmia, the authors should make exploratory statistics to compare with the other children (table 3). Of interest, there seems to be more patients with allergy and atopic disorders. Were alternate cause of anosmia sought in these patients (ie, nasal papulosis) ? Indeed, the criteria for long-COVID necessitate to exclude alternative diagnosis. Please discuss.

- The last part on patient voice is unusual. It is unfortunate that the patient did not use standardized questionnaire to assess the impact on the patients. Moreover, saying the age, sex & first name of a patient in a single center study is likely to breach anonymisation of the data. 

- The discussion does not discuss the data which are the prevalence of anosmia in kids; and the impact on their quality of life. Rather the authors discuss physiopathological hypothesis that have little to do with the current manuscript. 

Minor points 

- Error in the legend figure 1 : 13 not 10. This figure is not very readable. I suggest the author show an histogram of the median with individual points for duration and a different color for boy and for girl.

- Generally, the authors should give median and IQR25-75 and not range. 

- What % of the PARENTS have persistent symptoms related to COVID-19 ?

Author Response

Dear Editor and Reviewer, thank you very much for your efforts in reviewing our paper and improving it.
Please find below a point-by-point response to your comments.
Changes have been highlighted in yellow in the main manuscript

Buonsenso and colleagues present a retrospective study assessing the prevalence of anosmia among children < 18 consulting for persisting symptoms related to COVID-19. The authors report the data from 784 children and teenager from a single center in Italy and find 1.7% of reported persistent anosmia.

The results are interesting although similar data have been reported. Several points need to be considered.

Major point :

- The introduction focuses on Long-COVID in children, but it is not clear if the included patients were satisfying the criteria for long-COVID (for other symptoms than anosmia).

Thank you for the observation. This is a timely observation as the definition of Long Covid is not fully agreed worldwide. According to a recent DELPHI, any persisting symptoms lasting more than 12 weeks with a negative impact on daily life is compatible with long covid. More specifically, however, the 13 children complaining about smell disorders presented a broad spectrum of long lasting symptoms, developed after the Sars-CoV-2 acute infection, and affecting their wellbeing and return to social activities. So, their clinical presentation fit with the diagnosis of long-COVID.

- Another major point to discuss is that the population is a population (of parents) seeking medical attention after COVID-19. Since it is not a systematic analysis of infected patient the prevalence of anosmia cannot be generalized. Please discuss in the limitation.

Thank you, we clarified it in the discussion as follows “ For this reason, the prevalence of long lasting olfactory impairment among the pediatric population cannot be generalized”.

- In the abstract the authors state that anosmia is a COMMON long-term consequence of SARS-CoV-2 in adults. This is fairly rare, from 3-8% (10.3390/pathogens11020269). The introduction should contain what is already known on the subjet in adults and more importantly in children. The what is the gap of knowledge and then what the authors want to address. 

Since the beginning of the pandemic, the prevalence of post-COVID-19 anosmia in adults varies widely between studies and systematic studies conducted on pediatric population are still lacking, except for the one we cited in the discussion. Thank to your comment we changed the structure of the article and we added to the introduction the possible mechanisms of anosmia. We also highlighted the necessity to better understand the physiopathology of this problem and to find to most appropriate management.

- Even if there are few patients with anosmia, the authors should make exploratory statistics to compare with the other children (table 3). Of interest, there seems to be more patients with allergy and atopic disorders. Were alternate cause of anosmia sought in these patients (ie, nasal papulosis) ? Indeed, the criteria for long-COVID necessitate to exclude alternative diagnosis. Please discuss.

Our statistics consultant suggested that we could only make descriptive data because of the small number of patients with anosmia. However, if you can suggest us any tests we will try to perform them.

Moreover, The reviewer is completely right, the criteria for long-COVID necessitate to exclude the alternative diagnosis. In this specific group of patients, the clinical presentation of a broad spectrum of symptoms, including anosmia, persisting after the acute Sars-CoV-2 infection, was not suggesting us other underlying pathologies that could cause olfactory impairment. These problems are uncommon in pediatrics and it is unusual to find any alternavite diagnosis explaining this symptom. However, we have added this in the limitation section.

- The last part on patient voice is unusual. It is unfortunate that the patient did not use standardized questionnaire to assess the impact on the patients. Moreover, saying the age, sex & first name of a patient in a single center study is likely to breach anonymisation of the data.

We added a paragraph on patients’ voices to bring to light the difficulties they are facing in everyday life. We thought that getting insight in families’ views could be an original way to assess the impact of anosmia on their wellbeing.  However, in the future we will try to perform olfactory tests to obtain a more objective assessment. Also, Thank you for the comment, we deleted the names of the patients to preserve the anonymization of the data.

- The discussion does not discuss the data which are the prevalence of anosmia in kids; and the impact on their quality of life. Rather the authors discuss physiopathological hypothesis that have little to do with the current manuscript. 

Thank you for this observation. We just changed the structure of our article and in the discussion we analyzed the prevalence of anosmia among children reported in literature, our data and more important the impact on their families’ quality of life that emerged from our questionnaire.

Minor points 

- Error in the legend figure 1 : 13 not 10. This figure is not very readable. I suggest the author show an histogram of the median with individual points for duration and a different color for boy and for girl.

We corrected the error and changed the figure according to your suggestion.

- Generally, the authors should give median and IQR25-75 and not range. 

We corrected the table 3 with IQR.

- What % of the PARENTS have persistent symptoms related to COVID-19 ?

We only focused on children’s persistent symptoms. However, no parent of children with anosmia referred to suffer from persistent olfactory dysfunction.

Reviewer 2 Report

In this study the authors aimed to analyze the prevalence of long-lasting loss of smell among children who suffered from COVID-19 and to evaluate its severity, how it influenced their families' quality of life. The topic is interesting, however, several issues should be discussed in order to improve the quality of this manuscript.

Firstly, several language and stylistic errors should be corrected. Please explain all the abbreviations at first time they are used, including the abstract. Please use the proper names (COVID-19, not Covid; SARS-CoV-2 not SARS-Cov2) and be consisted throughout the manuscript.

Major issues:

1. Anosmia is a highly subjective disorder. In addition, children (in particular the youngest) may have problems with defining and reporting it.  Were there any olfactory testing performed in this group of patients?

2. There was one child with olfactory disfunction below 9 years of age. What age was the child?

3. Are there any data available on the short term olfactory disfunction in this group? It would be interesting to know, how many children with this problem improved spontanously in the short period observation.

4. The authors aimed to analyze the influence of the olfactory disfunction on the families' quality of life. How was this analyzed? What questionaires were used? Were they validated? In my opinion, giving data on how Sofia or Caterina felt is inappropriate in the scientific report.

5. What do you mean by "fully vaccinated" - does it refer to COVID-19 or all the recommended vaccinations?

6. Table 1. I think, that giving all the data in such a small group is not necessary. However, did you find any significant differences, eg. considering asthma? Please discuss.

7. 5% of patients were asyptomatic. As among inclusion criteria you mentioned "The child sought/needed primary or secondary medical care for COVID-19" - please explain why these asyptomatic patients required medical care?

Author Response

Dear Editor and Reviewer, thank you very much for your efforts in reviewing our paper and improving it.
Please find below a point-by-point response to your comments.
Changes have been highlighted in yellow in the main manuscript

In this study the authors aimed to analyze the prevalence of long-lasting loss of smell among children who suffered from COVID-19 and to evaluate its severity, how it influenced their families' quality of life. The topic is interesting, however, several issues should be discussed in order to improve the quality of this manuscript.

Firstly, several language and stylistic errors should be corrected. Please explain all the abbreviations at first time they are used, including the abstract. Please use the proper names (COVID-19, not Covid; SARS-CoV-2 not SARS-Cov2) and be consisted throughout the manuscript.

Major issues:

  1. Anosmia is a highly subjective disorder. In addition, children (in particular the youngest) may have problems with defining and reporting it.  Were there any olfactory testing performed in this group of patients?

Thank you for the observation. The reviewer is right, olfactory dysfunction is difficult to characterize especially in children. For this study, we used a qualitative research approach and we did not perform any olfactory tests.  However, we are currently trying to perform Sniff test in our patients so that in future we would like to make a more objective assessment of anosmia. We mentioned this as a limitation (Third, smell problems were defined according to a self-description of smell impairment by the patients, in the absence of a standardized diagnostic tool [19).

  1. There was one child with olfactory disfunction below 9 years of age. What age was the child?

The child in question was nine years old. We clarified it in the results section.

  1. Are there any data available on the short term olfactory disfunction in this group? It would be interesting to know, how many children with this problem improved spontaneously in the short period observation.

Thank you for the interesting question. Our data showed that none of our children improved spontaneously in the short period observation. For some of our patients, as we reported in figure 1, the olfactory impairment lasted a really long time.  

  1. The authors aimed to analyze the influence of the olfactory disfunction on the families' quality of life. How was this analyzed? What questionaires were used? Were they validated? In my opinion, giving data on how Sofia or Caterina felt is inappropriate in the scientific report.

We used a qualitative non validated questionnaire, which is reported in Table 2, with the aim of analyze families’ experiences and how their daily life has changed. Thank you for your comment, we deleted the names of the patients in order to preserve the anonymization of the data. We mentioned this in the study limitatinos-

  1. What do you mean by "fully vaccinated" - does it refer to COVID-19 or all the recommended vaccinations?

We referred only to COVID-19 vaccine. We clarified it in the results section.

  1. Table 1. I think, that giving all the data in such a small group is not necessary. However, did you find any significant differences, eg. considering asthma? Please discuss.

Thank you for the question. Due to the small number of patients with smell disorders, we could not perform statistical analyses, although allergies were more common in the anosmia group. As we answered to the first reviewer, the clinical presentation of a cluster of symptoms, including anosmia, persisting after the acute Sars-CoV-2 infection, was not suggesting us other underlying pathologies that could cause olfactory impairment.

  1. 5% of patients were asyptomatic. As among inclusion criteria you mentioned "The child sought/needed primary or secondary medical care for COVID-19" - please explain why these asyptomatic patients required medical care?

Thank you for this observation. During the highest peaks of Sars-CoV-2 infections, several children, who were exposed to the virus in household or school settings, went to the emergency department to do COVID-19 test and have a check on their clinical conditions.

Round 2

Reviewer 1 Report

The authors have addressed only part of the concerns.

1/ The long part of the discussion that was not relevant to this paper (anosmia pathophysiology) was moved to the introduction where it makes even less sense.

2/ The authors should ask their statistic advisor for the statistical test to use... Chi2 or Fisher's for qualitative; Student's or Mann-Whitney for quantititative data.. This was also asked by the other reviewer.

3/ The authors said that these patients had a broad range of symptoms. Please specify which ones in which patients (ie, in a suppl table). Indeed, this is important since persistent anosmia CAN be isolated.

Author Response

Thanks again gor your support, please find below a point-by-point response. Changes have been highlighted in the main manuscript.

1/ The long part of the discussion that was not relevant to this paper (anosmia pathophysiology) was moved to the introduction where it makes even less sense.

We modified the introduction only mentioning the main mechanisms involved in the pathogenesis of anosmia.

2/ The authors should ask their statistic advisor for the statistical test to use... Chi2 or Fisher's for qualitative; Student's or Mann-Whitney for quantititative data.. This was also asked by the other reviewer.

Thank you, we followed your suggestion and asked support to a local expert that we have mentioned in the acknowledgement section. We made a statistical analysis that showed that the distribution of age was significantly different between the anosmic and non anosmic patients. It also showed that the presence of altered taste, dyspnea at rest, dyspnea on exertion, chest pain, palpitations and joint pain was significantly different between the two groups of patients (p<0.05).

3/ The authors said that these patients had a broad range of symptoms. Please specify which ones in which patients (ie, in a suppl table). Indeed, this is important since persistent anosmia CAN be isolated.

Thank you for the observation, we added a table with the clinical presentation of the 13 patients. We visited only one patient with isolated anosmia, three patients with anosmia and dysgeusia.

Reviewer 2 Report

The revised version of the manuscript reads much better, most of the queries were answered. However, still the authos are inconsistent with tne nomenclature: please write SARS-CoV-2, not Sars, COVID-19, and not Covid 19, whcih also includes Long COVID.

Author Response

Thank you for appreciating our efforts. We corrected the nomenclature.